# Pharmacological Adherence Behavior Changes during COVID-19 Outbreak in a Portugal Patient Cohort

**DOI:** 10.3390/ijerph19031135

**Published:** 2022-01-20

**Authors:** Luís Midão, Marta Almada, Joana Carrilho, Rute Sampaio, Elísio Costa

**Affiliations:** 1Associate Laboratory i4HB-Institute for Health and Bioeconomy, Faculty of Pharmacy, University of Porto, 4050-313 Porto, Portugal; luismidao@gmail.com (L.M.); martassalmada@gmail.com (M.A.); 2UCIBIO–Applied Molecular Biosciences Unit, Faculty of Pharmacy, University of Porto, 4050-313 Porto, Portugal; 3Porto4Ageing-Competence Centre on Active and Healthy Ageing, University of Porto, 4050-313 Porto, Portugal; jcarrilho@reit.up.pt; 4Institute of Biomedical Sciences Abel Salazar, University of Porto, 4050-313 Porto, Portugal; 5CINTESIS-Center for Research in Health Technologies and Services, 4200-450 Porto, Portugal; rutesampaio@med.up.pt; 6Faculty of Medicine, University of Porto, 4200-450 Porto, Portugal

**Keywords:** adherence, COVID-19, impact, behavior, Portugal

## Abstract

Concerns, behaviours, and beliefs influence how people deal with COVID-19. Understanding the factors influencing adherence behaviour is of utmost importance to develop tailored interventions to increase adherence within this context. Hence, we aimed to understand how COVID-19 affected adherence behaviour in Portugal. A cross-sectional online survey was conducted between 1 March and 3 April 2021. Descriptive statistics were performed, as well as univariable and multivariable regression models. Of the 1202 participants, 476 who were taking at least one medication prescribed by the doctor were selected. Of these, 78.2% were female, and the mean age was 40.3 ± 17.9 years old. About 74.2% were classified as being highly adherent. During the pandemic, 8.2% of participants reported that their adherence improved, while 5.9% had worsened adherence results. Compared with being single, widowers were 3 times more prone to be less adherent (OR:3.390 [1.106–10.390], *p* = 0.033). Comorbid patients were 1.8 times (OR:1.824 [1.155–2.881], *p* = 0.010) more prone to be less adherent. Participants who reported that COVID-19 negatively impacted their adherence were 5.6 times more prone to be less adherent, compared with those who reported no changes (OR:5.576 [2.420–12.847], *p* < 0.001). None of the other variables showed to be significantly associated with pharmacological adherence.

## 1. Introduction

Coronavirus disease 19 (COVID-19) caused by the novel severe acute respiratory syndrome coronavirus 2 (SARS-CoV-2), was first identified in humans at the end of 2019, in the Chinese city of Wuhan [1]. The devastating effect of the SARS-CoV-2 has led to more than 5.23 million deaths and 264.37 million cases worldwide as of 3 December 2021 [2]. Among these, Portugal registered 1.15 million cases and 18,471 deaths [2], with the first confirmed case of COVID-19 on 2 March 2020, and 14 days later, the first associated death [3].

The COVID-19 pandemic impacted life in all its aspects [4]. The economic recession caused by COVID-19 had a huge effect on the labour market, led to the aggravation of inequalities and an increase in the number of people at risk of poverty, social exclusion, and reduced access to services [5,6]. From a social and psychological perspective, the pandemic and the restrictions imposed, such as social isolation and quarantine have had a tremendous impact on people’s mental health, having greatly increased anxiety, depression, and even suicide rates [7,8]. COVID-19 had also a remarkable impact on the health system, exacerbating already existing inequalities [9].

COVID-19 is often associated with a mild to severe respiratory disease, however, infection with this pathogen range from asymptomatic to critical cases [10,11]. Common symptoms include fatigue, dry cough, fever, headache, nausea, vomiting, dizziness, abdominal pain, anosmia and dysgeusia [12,13]. Anyone can be infected with COVID-19; however, some groups can be more prone to infection, and in case of an infection, are more likely to develop worst health outcomes. There is increasing evidence showing that patients with non-communicable diseases (NCDs) are at increased risk of contracting COVID-19 and of suffering worse health outcomes after infection [14,15,16]. NCDs had not been the focus of health services, since the outbreak of COVID-19 which is a critical flaw as they are responsible for 70–88% of deaths worldwide [15,17,18,19]. Also, this gap was intensified not only due to social isolation, but also due to mobility restrictions, the generalized fear of using health services, to repurposing of resources to patients infected with COVID-19, and the general difficulty in accessing care [16].

Medication persistence is the key for effective management of NCDs, and ensuring continuous access to medication is a prerequisite for proper medication adherence, which could be affected by disruptions in health services and social restrictions from the ongoing sanitary crisis [17]. Understanding the factors influencing adherence behaviour is of utmost importance to develop tailored interventions to increase it, which was already important, but became an urgent challenge at economic, healthcare, and social levels within the pandemic context. Hence, we aimed to understand how COVID-19 affected adherence behaviour, after one year of pandemic, in Portugal.

## 2. Materials and Methods

### 2.1. Data Collection

A cross-sectional online study was conducted between 1 March and 3 April 2021 through an online survey platform, LimeSurvey, an easy to use, free and open-source online survey application that enables users to develop and publish online surveys, and collect responses, without doing any programming. A link was created and posted on social media platforms [20] and disseminated through the general mailing list of the University of Porto, for students, alumni, teachers and non-teaching staff. More details are available on the completed Checklist for Reporting Results in Internet E-Surveys (CHERRIES) (Table A1) [21].

### 2.2. Measures

A researcher-made questionnaire was developed within a multidisciplinary team of experts on adherence (pharmacists, medical doctor, and health psychologist) to collect data about:■Sociodemographic: gender, age, marital status, and level of education.■Health status: chronic diseases, medication, changes in self-perceived health status before and during the pandemic.■Medication adherence: Medication Adherence Rating Scale (MARS-P9).■COVID-19 impact on medication adherence: changes in self-perceived medication adherence before and during the pandemic, changes on medication adherence due to the pandemic and reasons.■COVID-19 impact on adherence to healthy lifestyles: changes on adherence to a healthy diet and physical exercise.■COVID-19 perceptions and impact on daily life: feeling nervous, anxious, or lonely during the pandemic; difficulties falling asleep; started taking prescribed medication for anxiety, depression, or sleep; have tested positive or knowing someone who has; and concerns about being infected.

### 2.3. Exposure Variable

We defined adherence as the exposure variable, measured by 9 questions (MARS-P9). Answers were recorded on a 5-point Likert scale, and higher scores indicate higher adherence. Using the median value of the MARS-P9 data as a cut-off point [22], patients were classified as high adherers (score ≥ 37) or low adherers (score ≤ 36).

### 2.4. Statistical Analysis

Data was exported and analysed using IBM SPSS Statistics (Version 26.0. Armonk, NY, USA). Continuous variables were expressed as mean ± standard deviation (SD) and categorical variables were summarized as percentages. Unilevel and multilevel logistic regression was used to estimate the associations between adherence and outcomes. The effect size of association was presented as odds ratios (ORs) and 95% confidence intervals. *p*-value less than 0.05 was considered statistically significant.

### 2.5. Ethics Approval

The study was approved by the Ethics Commission of the Faculty of Pharmacy of the University of Porto, Portugal (N° 43-11-2020). Before completing the questionnaire, all participants provided electronic informed consent. 

## 3. Results

### 3.1. Sample Characteristics

Of the 1202 participants that answered to the survey, in this study, only those who were taking at least one medication prescribed by the doctor were selected, resulting in a total of 476 (Table 1). The questionnaire was created in a responsive way, so only those participants answered the questions related to adherence (MARS-P9). Of these, 78.2% were female, and the mean age was 40.3 ± 17.9 years old. Participants were mainly single (52.5%) and married (36.8%). One-third of participants had at least a BSc, while 39.9% held an MSc or PhD degree. 

Among participants, the most common conditions were cardiovascular diseases (angina, heart attack, heart failure, thrombosis, stroke, hypertension, high cholesterol, among others) with 21.0% of prevalence, followed by pulmonary diseases (chronic bronchitis, emphysema, allergies, and asthma, among others), with 20.2% and by endocrine diseases (diabetes, hypo, and hyperthyroidism) with 6.9%. More than 32% of participants had at least 2 different chronic conditions, while 10.5% were under polypharmacy. One year after the first case of infection in Portugal, 51 participants (10.7%) had already tested positive for COVID-19 and 451 (94.7%) knew someone who had tested positive.

### 3.2. Impact of COVID-19 on Medication Adherence

To better understand the impact of the pandemic context on medication adherence behaviour, participants were asked if COVID-19 had an impact on how they adhered to medication, and if so, in what way (positive or negative) this impact was, and what were the reasons for this (Table 2). Although 85.9% of the participants reported that no differences on their adherence were observed, 67 participants (14.1%) reported differences on their adherence due to COVID-19 pandemic. Of these, 39 reported improved adherence, mainly due to awareness of their own health status (*n* = 23), fear of COVID-19 which led them to improve their health status (*n* = 14), and more time for personal care (*n* = 8). On the other hand, 28 participants described declined adherence, mainly due to lack of support from family, neighbours, or friends (*n* = 11), avoid taking the medications (*n* = 6), and fear of leaving home (*n* = 4).

### 3.3. Factors Associated with COVID-19 Related Adherence

A multilevel univariable logistic regression model was firstly used to assess potential factors associated with adherence, and significant covariates (*p* < 0.05) were included in a final multilevel multivariable logistic regression model. Using a backward selection method, the presented final model was composed of only significant covariates (Table 3).

#### 3.3.1. Unilevel Analysis

From the sociodemographic characteristics included, only marital status showed to be significantly associated with adherence. Comparing with being single, widowers are 3.45 times less prone to be high adherers (0.289 [0.100–0.830]). Regarding the health status, being a comorbid patient makes individuals more (1.827 [1.192–2.800]) prone to be high adherers. Being under polypharmacy makes individuals less prone (0.527 [0.285–0.974]) to be high adherers. Individuals who reported that COVID-19 negatively impacted their adherence were also less prone to be high adherers (0.188 [0.085–0.416]). No significant association was found between high adherers and adherence to healthy lifestyles. Within the COVID-19 perceptions and impact on daily life, people who felt lonely, anxious, or sleepy, who started to take prescribed medication for it, and who had trouble falling asleep, were less prone to be high adherers ((0.564 [0.344–0.924]), (0.641 [0.424100–0.830]) and (0.622 [0.388–0.997], respectively).

#### 3.3.2. Multilevel Analysis

The results indicated that widowed (0.295 [0.096–0.904]), comorbid (0.548 [0.347–0.865]) and participants who reported that their adherence was negatively impacted by COVID-19 (0.179 [0.078–0.413]) were less prone to be high adherers.

## 4. Discussion

Medication adherence was used as the independent variable and 476 participants were included. Compared with the Portuguese average, there is an over-representation of women and highly educated individuals. It has already been described that women are more likely to participate in this type of online studies, as well as individuals with more education [23,24]. At the same time, this over-representation of highly educated individuals is certainly explained by how this questionnaire was disseminated, one of the biases of this study, as reported on CHERRIES.

There is scarce information on the prevalence of chronic diseases in the general population. Most studies focus on the older population, or comorbidities in people with certain diseases, making it difficult to compare our results with those in the literature. The most prevalent conditions in this study were cardiovascular (21.0%), pulmonary (20.2%) and endocrine (6.9%) diseases, all associated with a higher risk of mortality due to COVID-19 infection [25]. Cancer, cardiovascular, respiratory, endocrine and kidney diseases are related to an increased risk of complications from COVID-19. The existence of more than one comorbidity is associated with an even greater risk of a worse prognosis [26,27,28,29]. We found that almost one third of participants had more than 1 chronic disease, which is in accordance with the prevalence reported previously among adults [30,31,32].

Polypharmacy, commonly defined as taking 5 or more different medications per day, can impact the health of patients [33]. It is already known that polypharmacy in patients infected with COVID-19 is related to worse health outcomes [34]. Although it is a more prevalent condition in the ageing population, it is something that cuts across all age groups. However, little information exists in large-scale studies worldwide, with the adult population. In this study, a prevalence of polypharmacy of 10.5% was found, a value similar to studies from Poland and New Zealand, which recorded 11.7% and 9.9% of polymedicated individuals, respectively [35,36].

As of 3 April 2021, the last day this survey was collecting data, 823,335 people in Portugal had already been infected by SARS-CoV-2, which represented 7.95% of the Portuguese population [37]. On this date, Portugal was one of the European Union countries with the highest percentage of COVID-19 infection. Worse than Portugal was Lithuania, Estonia, Luxembourg, Slovenia, Slovakia and the Czech Republic, with percentages between 8.1 and 14.5% [38]. In this study, until the same day, 10.5% of the participants had already been infected. The northern region of Portugal was where the first cases of COVID-19 began, and since the beginning of the pandemic, it is one of the areas in Portugal with a higher prevalence of cases. Considering the survey’s dissemination strategy, it is also clear why the average of infected participants is slightly higher than the national average, since it has more people from the north region of Portugal.

The COVID-19 pandemic had an impact on the self-perceived behaviour of adherence to therapy. In every 7 people, 1 reported that their adherence was changed due to the pandemic. Of these, nearly 60% reported improvements while more than 40% reported worsening in the way they adhered to their treatment regimen. The most mentioned reason among participants for improving adherence to therapy was being more aware of their health status (59.0%). There are already studies that show a relationship between being more aware of the health status, with the promotion of healthy lifestyles in chronic patients [39,40]. Another reason mentioned was wanting to improve their health status for fear of being infected (35.9%). Fear can have two opposite effects. On the one hand, it can promote healthy lifestyle habits and be a motivator to control danger in certain situations. Fear of being infected was indeed one of the reasons that led participants in another study to use and adhere more to herbal medicine to prevent or control the symptoms of SARS-CoV-2 infection [41]. Having more time for personal care and feeling capable to do it, were other two reasons why people adhered more (20.5% and 15.4%, respectively). A study from Mexico showed that during the COVID-19 outbreak, 60–80% of people adopted self-care behaviours [42]. Social support plays a fundamental role in people’s well-being. In difficult times, such as in times of pandemic, social support works as an element of protection that helps individuals to cope with stressful situations more efficiently [43,44,45]. Having this support from family, neighbours or friends was another reason why people adhered more (12.8%). On the contrary, the lack of this social support was the most mentioned reason, among the participants who reported having worse adherence due to the pandemic (39.3%). Fear, on the other hand, can be paralyzing among more vulnerable individuals, preventing the adoption of these habits [46]. Fear of leaving the house or of going to a pharmacy were also reported to decrease adherence (14.3% and 3.6%, respectively). During the pandemic, access to medical appointments and contact with physicians was reduced, hence pharmacists played a pivotal role in patient education [47]. This fear of leaving the house and going to the pharmacy not only had a negative effect on adherence but also on this contact, which further impacts health. This health literacy would be of value since other reasons why people adhered less were avoiding taking the medicines (21.4%) and being afraid of secondary effects (3.6%). A small proportion (7.6%) reported feeling less able to take care of themselves, and the impossibility of moving by their own means was the last reason why people adhered less (3.6%). Although costs of medication were already reported as barriers to medication adherence [48,49], in this study, no participant reported lack of money as a reason for decreased adherence.

For most participants, the way they perceived their health before and during the pandemic did not change. However, 25.6% of participants reported negative changes in their self-perception of health. This decrease in self-perceived health has also been reported in another study, in Germany, which was associated with stress [50]. Poorer self-perception of health during the pandemic was also related to fear of being infected, psychological discomfort and loneliness [51,52]. Only 3.8% of participants reported positive changes in their self-perceived health. A possible explanation for this is that a part of the participants took advantage of the pandemic, quarantine, and isolation to improve their habits, either of physical exercise (20.8%) or healthy eating (21.6%). This change could have been more significant, but most of the population reported high levels of health both before and during the pandemic. The WHO, along with social isolation and the obligation to stay at home, anticipated an overall decrease in physical exercise. As previously reported, many participants in this study (43.9%), started to exercise less or stopped exercising completely [53,54]. However, in our study, as well as several others in Italy, Spain and Poland, a significant percentage of participants started to exercise or started to exercise more [53,54,55]. Regarding adherence to a healthy diet, studies reported improvements in 28–34% of participants, while in this study we reported that 21.6% of participants adhered to a healthier diet [54,55].

The pandemic also had an impact on mental, physical and psychological well-being, on emotional disorders as well as on the quality of sleep [56]. In a study of nearly 500 participants from northern Italy, 43% reported symptoms of insomnia, a value in agreement to the 48% of participants in this study that reported having trouble falling asleep [54]. A systematic review that included 24 studies showed that loneliness was related to symptoms of mental health and sleep problems in the adult population during the COVID-19 pandemic [57]. In our study, almost 72% of participants also reported feeling lonely, nervous, or anxious during these times. It is also important to highlight that 22.1% of the participants reported having started taking medication prescribed by the doctor to control the increase in anxiety, difficulty falling asleep or depression due to COVID-19, also known as coronasomnia [58]. Out of 5 people, 4 were concerned about being infected with COVID-19. These concerns are already known to harm mental health, increasing stress and anxiety [59].

Studying the association between adherence and marital status, we found that widowed participants were less prone to be high adherers (0.295 [0.096–0.904]), compared with the single participants. The contributions of having a spouse range from the social, emotional, financial, and even assistance in the adherence process [60,61]. Compared to those who had none or only 1 chronic disease, comorbid patients were also more prone to be high adherers (0.548 [0.347–0.865]). The existence of comorbidities often leads to the prescription of several medications, thus leading to polypharmacy, which can reduce adherence to it [62]. In older adults, who are often comorbid patients, the non-adherence rate ranges from 43–100%, which is highly associated with disease progression, treatment failure, hospitalizations and even death [63]. Finally, and as expected, participants who reported that their adherence was negatively impacted by COVID-19 (0.179 [0.078–0.413]) were less prone to be high adherers. Of all the other variables studied, none were listed as being significantly associated with adherence, including adherence to a healthy diet and exercise.

### Limitations

This study, as already mentioned, has some limitations. The way this questionnaire was disseminated, and the fact that it was done online, leads to bias. It is known that people who participate in this type of studies tend to be younger, motivated, educated and healthier than those who refuse to respond or who are unable to participate, so older people, participants with comorbidities, and with a lower degree of education may have been excluded. Our respondents are highly educated people, which is not necessarily representative of the general population. However, if in this population we have already seen that the pandemic has affected the behavior of adherence to therapy, in the general population this effect is certainly greater, so future studies in this area should consider this.

## 5. Conclusions

To our knowledge, this is the first work that shows the impacts of the pandemic on the behavior of adherence to therapy. Non-adherence was already a problem before the pandemic, and it has been exacerbated with COVID-19 and all the associated limitations. Awareness of the evolution of behaviours, concerns, and attitudes in unusual and novel situations, such as the case of this pandemic, is crucial to understand its outcomes in a country. Moreover, this knowledge is very important for the development of strategies to mitigate the effects of the pandemic, and all its consequences.

## Figures and Tables

**Table 1 ijerph-19-01135-t001:** Summary of the characteristics of the population included in this study.

Variables	Category	Frequency	%
Gender	Female	372	78.2
	Male	104	21.8
Age		40.3 ± 17.9 years	
Marital status	Single	250	52.5
	Married	175	36.8
	Divorced	36	7.6
	Widowed	15	3.2
Education	High School	129	27.1
	Bachelor	157	33.0
	Master or PhD	190	39.9
Type of chronic disease	Cardiovascular	100	21.0
	Pulmonary	96	20.2
	Endocrine	83	6.9
	Gastrointestinal	27	5.7
	Joint, Muscle and Bone	24	5.0
	Skin	9	1.9
	Cancer	11	0.9
	Kidney	3	0.6
	Genetic	3	0.6
	Pain	2	0.4
	Infectious	2	0.4
	Eye	0	0.0
	Neurodegenerative	0	0.0
	Psychological	0	0.0
	Other	3	0.6
Number of chronic diseases	≥2	153	32.1
	<2	323	67.9
Number of different medications taken per day	≥5	50	10.5
	<5	426	89.5

**Table 2 ijerph-19-01135-t002:** Impact of the COVID-19 pandemic situation on the adherence behaviour.

			Reason	Frequency	%
COVID-19 Impacted Adherence	Yes 67 (14.1%)	Improved 39 (58.2%)	Awareness of health status	23	59.0
			Improve health status for fear of COVID-19	14	35.9
			More time for personal care	8	20.5
			Feeling more able to take care of self	6	15.4
			Support from family, neighbours, or friends	5	12.8
		Declined 28 (41.8%)	Lack of support from family, neighbours, or friends	11	39.3
			Avoid taking the medications	6	21.4
			Fear of leaving home	4	14.3
			Feeling less able to take care of self	2	7.1
			Fear of going to the pharmacy	1	3.6
			Impossibility of moving by own means	1	3.6
			Afraid of secondary effects	1	3.6
			Economic reasons	0	0.0
	No 358 (75.2%)				
	Indifferent 51 (10.7%)				

**Table 3 ijerph-19-01135-t003:** Association of sociodemographic, health status, COVID-19 impact on medication adherence, COVID-19 impact on adherence to healthy lifestyles, and COVID-19 perceptions and impact on daily life with adherence: univariate and multivariate analysis.

		*n*	*n* Low Adherers (%)	*n* High Adherers (%)	Unilevel Analysis			Multilevel Analysis		
		476	123 (25.8)	353 (74.2)	OR	CI 95	*p*	OR	CI 95	*p*
Sociodemographic	Gender									
	Female	372	97 (78.9)	275 (77.9)	1	-	-	-	-	-
	Male	104	26 (21.1)	78 (22.1)	0.945	0.572–1.561	0.825	-	-	-
	Age									
	18–24 years	152	39 (31.7)	113 (32.0)	1	-	-	-	-	-
	25–34 years	57	11 (8.9)	46 (13.0)	1.443	0.679–3.067	0.339	-	-	-
	35–44 years	65	14 (11.4)	51 (14.4)	1.257	0.627–2.522	0.518	-	-	-
	45–54 years	78	23 (18.7)	55 (15.6)	0.825	0.449–1.518	0.536	-	-	-
	≥55 years	124	36 (29.3)	88 (24.9)	0.844	0.495–1.438	0.531	-	-	-
	Marital Status									
	Single	250	62 (50.4)	188 (53.3)	1	-	-	1	-	-
	Married	175	42 (34.1)	133 (37.7)	1.044	0.665–1.640	0.850	1.042	0.614–1.692	0.868
	Divorced	36	11 (8.9)	25 (7.1)	0.750	0.348–1.614	0.460	0.719	0.317–1.633	0.430
	Widowed	15	8 (6.5)	7 (2.0)	0.289	0.100–0.830	0.021	0.295	0.096–0.904	0.033
	Education									
	Until High School	129	32 (26.0)	97 (27.5)	1	-	-	-	-	-
	Degree	157	51 (41.5)	106 (30.0)	0.686	0.407–1.156	0.156	-	-	-
	Master or PhD	190	40 (32.5)	150 (42.5)	1.237	0.727–2.105	0.432	-	-	-
Health status	Number of chronic diseases									
	≥2	153	52 (42.3)	101 (28.6)	1	-	-	1	-	-
	<2	323	71 (57.7)	252 (71.4)	1.827	1.192–2.800	0.006	0.548	0.347–0.865	0.010
	Polypharmacy									
	No	426	104 (84.6)	322 (91.2)	1	-	-	-	-	-
	Yes	50	19 (15.4)	31 (8.8)	0.527	0.285–0.974	0.041	-	-	-
	Self-perceived health changed due to COVID-19									
	No	336	83 (67.5)	253 (71.7)	1	-	-	-	-	-
	Negatively	122	36 (29.3)	86 (24.4)	0.784	0.493–1.245	0.301	-	-	-
	Positively	18	4 (3.3)	14 (4.0)	1.148	0.367–3.595	0.812	-	-	-
COVID-19 impact on medication adherence	COVID-19 impacted adherence									
	No	409	92 (74.8)	317 (89.8)	1	-	-	1	-	-
	Negatively	28	17 (13.8)	11 (3.1)	0.188	0.085–0.416	<0.001	0.179	0.078–0.413	<0.001
	Positively	39	14 (11.4)	25 (7.1)	0.518	0.258–1.039	0.064	0.514	0.247–1.070	0.075
COVID-19 impact on adherence to healthy lifestyles	Changes in adherence to healthy diet									
	No changes	290	68 (55.3)	222 (62.9)	1	-	-	-	-	-
	Yes, for a less healthy diet	83	25 (20.3)	58 (16.4)	0.711	0.413–1.224	0.217	-	-	-
	Yes, for a healthier diet	103	30 (24.4)	73 (20.7)	0.745	0.449–1.236	0.254	-	-	-
	Changes in adherence to physical exercise									
	No changes	168	38 (30.9)	130 (36.8)	1	-	-	-	-	-
	Stopped or started to exercise less	209	54 (43.9)	155 (43.9)	0.839	0.521–1.352	0.119	-	-	-
	Started to practice (more) exercise	99	31 (25.2)	68 (19.3)	0.641	0.367–1.122	0.470	-	-	-
COVID-19 perceptions and impact on daily life	Felt lonely, anxious, or nervous									
	No	135	25 (20.3)	110 (31.2)	1	-	-	-	-	-
	Yes	341	98 (79.7)	243 (68.8)	0.564	0.344–0.924	0.023	-	-	-
	Trouble falling asleep									
	No	248	54 (43.9)	194 (55.0)	1	-	-	-	-	-
	Yes	228	69 (56.1)	159 (45.0)	0.641	0.424–0.971	0.036	-	-	-
	During the context of the pandemic COVID-19 started taking medication for anxiety, depression or difficulty falling asleep by prescription									
	No	371	88 (71.5)	283 (80.2)	1	-	-	-	-	-
	Yes	105	35 (28.5)	70 (19.8)	0.622	0.388–0.997	0.049	-	-	-
	Concerned being infected with COVID-19 *									
	No	83	23 (21.1)	60 (19.0)	1	-	-	-	-	-
	Yes	342	86 (78.9)	256 (81.0)	0.876	0.510–1.505	0.632	-	-	-

* 51 participants were already infected, and therefore did not answer to this question.

## Data Availability

The data presented in this study are available on request from the corresponding author.

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
