# Peer review of "Pharmacological Adherence Behavior Changes during COVID-19 Outbreak in a Portugal Patient Cohort"

_ijerph, 2022, doi:10.3390/ijerph19031135_

Round 1

Reviewer 1 Report

There is a very important and relevant message in this paper " Adherence Behavior Change During COVID-19 Outbreak in Portugal ".
How people cope with COVID-19 depends on their concerns, behaviors, and beliefs. In order to develop tailored interventions to increase adherence, it is of utmost importance to identify the factors that influence adherence behavior. 
There is a need to explain in more detail the conduct of the surveys, the processing of the results, and the significance of the results of the surveys in the article. Language and style should be appropriate. The spelling needs to be revised.  

Author Response

Comment 1: There is a need to explain in more detail the conduct of the surveys, the processing of the results, and the significance of the results of the surveys in the article. Language and style should be appropriate. The spelling needs to be revised.  

Answer 1: First, we would like to thank you for your completely pertinent comment. The article has been revised, both linguistically and grammatically, and we explored further the points you highlighted.

Reviewer 2 Report

I read with great interest the work of Midao et al, since the topic is of great interest and it involves this contemporary event that is continuously changing our life. Their study on an European population group could be also considered and compared to other similar population groups to find optimal solutions of behavior changing awareness.

The article is well written with minor spelling which should be corrected such as:

Line 171 "to the response, to the disease"

Line 237 "to to"

Also the authors should consider the following:

Line 53 " some groups can be more prone to infection or adverse reactions" It is not what adverse reactions the authors are referring to, or to what in particular.

The authors should also consider in adding the following references since also medical personnel may be affected in various ways by the pandemic, and pharmacist may help in surpassing the polypharmacy programs by other using different programs to aid patients.

  • Doica IP, Florescu DN, Oancea CN, Turcu-Stiolica A, Subtirelu MS, Dumitra G, Rogoveanu I, Gheonea DI, Ungureanu BS.  Telemedicine Chronic Viral Hepatitis C Treatment during the Lockdown Period in Romania: A Pilot Study. Int J Environ Res Public Health. 2021 Apr 1;18(7):3694. doi: 10.3390/ijerph18073694.

Authors should also consider in shortening the discussions section and mention some of the study limitations.

Author Response

Comment 1: Line 171 "to the response, to the disease"

Answer 1: We would like to thank the reviewer for all the suggestions, that were helpful to improve the quality of our paper. We have addressed this issue.

Comment 2: Line 237 "to to"

Answer 2: Already corrected.

Comment 3: Line 53 " some groups can be more prone to infection or adverse reactions" It is not what adverse reactions the authors are referring to, or to what in particular.

Answer 3: With that particular sentence, we wanted to raise the awareness that people within some groups were more prone to be infected, and in case of an infection, were also more likely to develop worst health outcomes. Although, the sentence was not that clear, and we had changed it, so we thank you for your input.

Comment 4: The authors should also consider in adding the following references since also medical personnel may be affected in various ways by the pandemic, and pharmacist may help in surpassing the polypharmacy programs by other using different programs to aid patients.

  • Doica IP, Florescu DN, Oancea CN, Turcu-Stiolica A, Subtirelu MS, Dumitra G, Rogoveanu I, Gheonea DI, Ungureanu BS.  Telemedicine Chronic Viral Hepatitis C Treatment during the Lockdown Period in Romania: A Pilot Study. Int J Environ Res Public Health. 2021 Apr 1;18(7):3694. doi: 10.3390/ijerph18073694.

Answer 4: We have added it to our manuscript. Thanks for poiting this interesting article.

Comment 5: Authors should also consider in shortening the discussions section and mention some of the study limitations.

Answer 5: We also considered this comment, and addressed it already, thank you for raising this point.

Reviewer 3 Report

Very interesting and topical topic for potential readers of this journal.

Some comments "in favor" of improving the manuscript:

.- Title. Maybe change by: “Pharmacological adherence behavior change during COVID-19 outbreak in a portugal patient cohort”.

.- Introduction. Maybe shorten. The first two paragraphs describe well-known information.

.- Material and methods. Maybe briefly explain the Limesurvey tool.

Some doubts arise about the study sample. Could be a selection bias when recruiting patients electronically? That is, could younger and more motivated patients (adherents of treatment) have signed up for this study?

Why did you select patients who were taking at least one medication and did not set a higher cut-off point?

.- Results. 409 patients (86%) consider that the pandemic has not influenced their pharmacological adherence behavior. Very striking positive result that should be included in the discussion and interpreted as perhaps younger patients with fewer medications are more compliant. Likewise, those more familiar with new technologies and more compliant could be more motivated to answer the electronic survey.

.- Table 1. How many had 2 diseases, 3 or more diseases? It could be an interesting piece of information, as well as relate it to the number of drugs.

.- Table 3. Several data are striking: Only 30% of the sample older than 55 years (from 65 years is when patients take more medications), 84.6% of the sample were from the non-polymedicated group.

.- Discussion. Too long. It is a descriptive of other studies. It is necessary to correlate the results of the study with other works. Likewise, a section on study limitations is missing.

.- Conclusions. They result too general and pretentious. They do not meet the objective of the study that is exposed in the last sentence of the introduction section (to know how the COVID-19 pandemic has affected compliance behavior one year later)

.- Bibliography. It´s wide and very up-to-date. 73 citations, the majority being very recent (2020-2021).

Major changes could be necessary.

Perhaps it could recovert as a Scientific Letter.

Author Response

Comment 1: Title. Maybe change by: “Pharmacological adherence behavior change during COVID-19 outbreak in a Portugal patient cohort”.

Answer 1: We would like to thank the reviewer for this valuable feedback. In fact, changing the title according to your suggestion, makes it more accurate.

Comment 2: Introduction. Maybe shorten. The first two paragraphs describe well-known information.

Answer 2: We have addressed it and decreased the length of the introduction.

Comment 3. Material and methods. Maybe briefly explain the Limesurvey tool.

Answer 3: We thank you for this comment, which helped us to make it clearer to the reader what is Limesurvey

Comment 4: Some doubts arise about the study sample. Could be a selection bias when recruiting patients electronically? That is, could younger and more motivated patients (adherents of treatment) have signed up for this study?

Answer 4: The method of recruiting participants led to some bias, as mentioned both in the discussion and in the CHERRIES table. In fact, we should give more emphasis to this, and other limitations of the study, so, as suggested by the reviewer, and we are grateful for the constructive criticism, a final paragraph with the limitations of the study was added.

Comment 5: Why did you select patients who were taking at least one medication and did not set a higher cut-off point?

Answer 5: The questionnaire was created in a responsive way. Only participants who took at least one medication prescribed by their physician answered the questions related to adherence (MARS-P9). Adherence being our main variable, we only selected the participants who answered these questions. However, with your comment, we realized that this is not obvious to the reader, and we have added that explanation. Thanks for the feedback.

Comment 6: Table 1. How many had 2 diseases, 3 or more diseases? It could be an interesting piece of information, as well as relate it to the number of drugs.

Answer 6: Thanks in advance for this comment. In fact, we did an analysis to understand the relationship between the number of diseases and the number of drugs taken. What we saw was that, as we expected and as has been reported countless times, people with more diseases took more medications. As it was not the focus of our work, and considering that our work was already quite extensive, this information, that we did not interpret as essential, was not included in the final version of the article.

Comment 7: Table 3. Several data are striking: Only 30% of the sample older than 55 years (from 65 years is when patients take more medications), 84.6% of the sample were from the non-polymedicated group.

Answer 7: As mentioned in the limitations of the study, and given the form of data collection, the age distribution of the sample is not representative of the general population. Although, overall, almost 85% of the population is not polymedicated, looking at the 55+ population, about 24% is polymedicated. If we were to divide the participants by age 65 and over, the figure would be even higher, which is in line with published European work that shows that around 33% of people aged 65 and over in Europe are on polypharmacy.

Comment 8: Discussion. Too long. It is a descriptive of other studies. It is necessary to correlate the results of the study with other works. Likewise, a section on study limitations is missing.

Answer 8: We appreciate your pertinent comment. According to it, we altered the discussion and reduced its length.

Round 2

Reviewer 3 Report

The suggested changes were positively perceived, except "Comment 8: Discussion. Too long. It is a descriptive of other studies. It is necessary to correlate the results of the study with other works."

Author Response

We would like to thank the reviewer once again for the pertinent and constructive feedback. We have addressed the issue, decreased the extension of the discussion section, and correlated our results with others previously described in the points it was missing.